# Analysis of Memory Organization for Dynamic Neural Networks

## Abstract

An increasing number of neural memory networks have been developed, leading to the need for a systematic approach to analyze and compare their underlying memory capabilities. Thus, in this paper, we propose a taxonomy for four popular dynamic models: vanilla recurrent neural network, long short-term memory, neural stack and neural RAM and their variants. Based on this taxonomy, we create a framework to analyze memory organization and then compare these network architectures. This analysis elucidates how different mapping functions capture the information in the past of the input, and helps to open the dynamic neural network black box from the perspective of memory usage. Four representative tasks that would fit optimally the characteristics of each memory network are carefully selected to show each network's expressive power. We also discuss how to use this taxonomy to help users select the most parsimonious type of memory network for a specific task. Two natural language processing applications are used to evaluate the methodology in a realistic setting.

## 1 Introduction

Memory is crucial for both cognitive science and machine learning in applications that require the use of past information such as sequence learning problems. It is fair to say that recurrent neural networks are the most popular memory networks nowadays. Elman and Jordan proposed the first classic version of the recurrent neural network (RNN) Elman (1990)Jordan (1986). which is often referred to as vanilla RNNs (vRNNs) currently. However, since the memory of vRNN is internal memory encoded in hidden states, vRNN is difficult to implement in practice: if the temporal dependency relationship of the time series is complex and long, the number of hidden units has to be tremendously increased to fully capture useful past information. The long range dependence problem is one common example of this difficultyTrinh et al. (2018)Belletti et al. (2018)Bengio et al. (1994).

Many dynamic neural networks with new memory structures have recently emerged as improvements of the vRNN architecture. (The terms recurrent, dynamic, and memory neural network are used interchangeably in this paper.) Some of these networks adopt internal memory; some adopt external memory; some adopt logic gates; while others adopt an attention mechanism. As expected, all these methods have advantages for some specific tasks, but selecting or designing an optimal memory model for a new task is very difficult unless we have a clear understanding of how their memory helps solve problem at hand. This is not an easy issue because 'memory' is a very abstract concept and the specific memory requirements for a specific task are implicit, which means that quantitatively conceptualizing and analyzing memory is a very hard problem. Cognitive scientists have defined many different types of memory, which shows the richness of the concept. In engineering, there are only a few engineering quantifiers of memory such as memory depth and memory resolution, but they are not enough for the ever-growing applications of machine learning.

Thus, the main goal of this paper is to investigate the characteristics of memory implemented by each architecture and what information these networks can extract from the input time series. We illustrate the role and characteristics of mapping functions and describe how memory is implemented of each memory network. We also classify popular memory networks into four types and find the essential relationships amongst these classes. The results are summarized in a rigorous taxonomy, which serves as a simple way to quantify the capabilities of extracting and using past information of each class. A secondary goal is to employ the knowledge gained from the different characteristics of

these memory structures to help users select the most parsimonious type of memory network given the problem at hand. We achieve this goal by connecting models' relative expressive power to the memory requirements of different tasks and use four synthetic tasks to exemplify each task type. Moreover, two natural language processing applications are used to evaluate the methodology in a realistic setting.

## 2 RELATED WORK

Among the abundant recurrent network literature, minimal attention has been given to understanding and analysis of their memory characteristics. Omlin & Giles (1996) discussed how vRNNs behaves like deterministic finite-state automata. Gers & Schmidhuber (2001)Rodriguez (2001)Schmidhuber et al. (2002) compared the performance of long short-term memory (LSTM) Hochreiter & Schmidhuber (1997) and vRNN on some context-free/sensitive language. Collins et al. (2016) studied the capacity of recurrent nets and how difficult they are to train. Karpathy et al. (2015) visualized long-term interactions and representations learned by recurrent networks. Greff et al. (2017) empirically studied the importance of various computational components of LSTM. Jozefowicz et al. (2015) evaluated a variety of recurrent neural network architectures and attempted to find the best one. Chung et al. (2014) compared gated recurrent unit (GRU) networks Cho et al. (2014) to LSTM. Yogatama et al. (2018) tested and compared performances of sequential, random access, and stack memory architectures on language modeling dataset. These works assessed network performance based on the output error. By contrast, our work focuses on how these networks encode information to solve a problem.

## 3 A MEMORY NETWORK TAXONOMY

In this section, we organize these popular dynamical networks according to their underlying memory structures. We classify them into four classes as shown in Fig.1, i.e., vRNN⊆ LSTM⊆neural stack⊆neural RAM. Some classes are named after a typical model. For example, the vRNN class also includes IRNN Le et al. (2015) and the highway network Srivastava et al. (2015), and the LSTM class also includes GRU Cho et al. (2014) and the peephole network Gers & Schmidhuber (2000). The neural stack class includes the architecture in Sun (1993); Sun et al. (2017); Joulin & Mikolov (2015); Grefenstette et al. (2015), and the neural RAM class includes NTM Graves et al. (2014), DNC Graves et al. (2016), enhanced LSTM Graves (2013) and Weston et al. (2014).

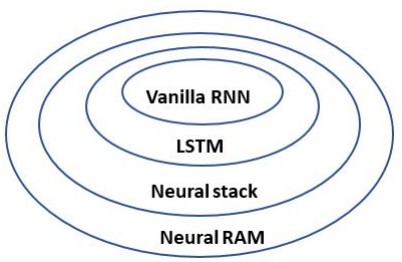

Figure 1: Memory Network Taxonomy

vRNN's internal memory has a lot of limitations since all the useful information is compressed in the hidden states that are continuously changing with the input samples: the past contents and their time information are mixed together, i.e., a certain event can not be extracted without interference, the hidden state has to be updated at every time steps, the long range dependency problem makes it hard to train, etc. These problems make vRNN unpractical. Compared to vRNN, the type of memory of LSTM, neural stack and neural RAM is external. LSTM introduced an external memory slot and the corresponding gate mechanism to address the long range dependency problem. With this external memory slot, a useful event in the past can be extracted without interference. However, if there are multiple useful events, they have to be compressed as a compounded event and accessed as a whole. The specific content of each event and their temporal order can not be captured. Neural stack and neural RAM addressed this problem by introducing multiple-slot external memory, where multiple events can be stored and their temporal order information is also retained. The difference between neural RAM and neural stack is the contents of neural RAM can be accessed randomly.

The different memory organizations give these networks different expressive power. Actually one class of network architectures can be seen as a special case of another class as shown in Fig.1. The memory types are summarized in Table.1. In next section, we will analyze their underlying structures in details and showing their essential relationships.

Table 1: Memory types of networks

| Networks | Memory type | |
|---|---|---|
| vRNN | state memory | internal memory |
| LSTM | memory of a single compounded event at a time | external memory |
| Neural stack | memory of multiple events, information of each event should be used sequentially, only one event is accessible at each time step | external memory |
| Neural RAM | memory of multiple events, all are accessible at each time step, no restriction on how many times they are used | external memory |

## 4 MEMORY STRUCTURE ANALYSIS

In this section, we analyze the memory structures of four popular RNNs: vRNN, LSTM, neural stack and neural RAM. Attention is paid to how their underlying memory organizations lead to different features and expressive power.

### 4.1 VRNN

The vRNN network Jordan (1986) is composed of three layers: input, hidden recurrent and output. In addition to the feedforward connections, a feedback connection from the hidden layer to itself exists. The architecture is shown in Fig.2(a). The dynamics of the hidden layer can be written as

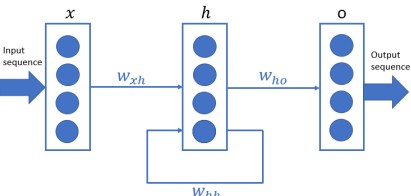

(a) Network architecture

$$\mathbf{h}_t = f(\mathbf{w}_{xh}^{\mathrm{T}}\mathbf{x}_t + \mathbf{w}_{hh}^{\mathrm{T}}\mathbf{h}_{t-1} + \mathbf{b}_h), \quad (1)$$

$$\mathbf{o}_t = f(\mathbf{w}_{ho}^{\mathrm{T}}\mathbf{h}_t + \mathbf{b}_o), \quad (2)$$

where $\mathbf{x}_t$, $\mathbf{h}_t$ and $\mathbf{o}_t$ are the input, hidden state and output vector at time $t$. We use $\mathbf{w}$ and $\mathbf{b}$ to represent the weight and bias, and $f(x)$ is the nonlinear activation function.

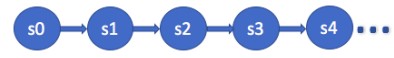

(b) Memory visualization

Figure 2: vRNN

**A vRNN induces memory by encoding the past information in its hidden state units $\mathbf{h}_t$, which are continuously changing and forced to be used every time; thus, the memory of the vRNN is called state memory or internal memory.** Fig.2(b) shows the state transition diagram of a vRNN, where $s0$, $s1$, ..., $s4$ represent the state at times $t_0$, $t_1$, ..., $t_4$, respectively. The arrows show the dependency of the variables. Here, state $s1$ is decided by only $s0$, $s2$ is decided by only $s1$ and so on. (All the memory visualization figures in this paper ignore the current input.)

As the number of hidden units is limited in practice, a compromise between memory depth and memory resolution always exists in the vRNN De Vries & Principe (1992). For long memory depth sequences, a vRNN requires a very large number of hidden units to achieve acceptable accuracy. If the sequences are composed of symbols or discrete numbers, the limitation of vRNN is even easier to see: the process can be viewed from a Markov transition model perspective. Specifically, the vRNN attempts to learn a first-order Markov transition model (with transition probability 1), where the current state is decided by only current input and the state in the previous step. Thus, since the state space is not very large (the number of states is less than the size of input symbols' alphabet) for first-order Markov sequences, the vRNN always performs well. However, for higher-order Markov sequences or sequences that do not satisfy the Markov property, the vRNN still attempts to build a first-order Markov state model, which results in a very large state space (several old states must be combined into a new state). The compromise between memory depth and memory resolution (which are related to the number and temporal resolution of the states) would make vRNN not suitable for these kinds of sequences. **Hence, vRNN is only suitable for lower-order Markov chains.**

### 4.2 LSTM

LSTM was proposed to address the vanishing gradient problem of vRNN. **It was the first memory network that had to address the problem of how to store and retrieve information from an external source to improve performance (through gating networks).** In this section, we analyze how LSTM provides greater flexibility from the perspective of memory usage.

In contrast to vRNN, in the classic LSTM shown in Fig.3(a), the feedback connection of the hidden layer must pass through external memory $\mathbf{m}_t$,

$$\mathbf{c}_t = f(\mathbf{w}_{hc}^{\mathrm{T}}\mathbf{h}_t + \mathbf{b}_c), \qquad (3)$$

$$\mathbf{m}_t = g_{i,t}\mathbf{c}_t + g_{f,t}\mathbf{m}_{t-1}, \qquad (4)$$

$$\mathbf{r}_t = \mathbf{m}_t, \qquad (5)$$

$$\mathbf{h}_t = f(\mathbf{w}_{xh}^{\mathrm{T}}\mathbf{x}_t + \mathbf{w}_{rh}^{\mathrm{T}}g_{o,t}\mathbf{r}_{t-1} + \mathbf{b}_h), \qquad (6)$$

where $\mathbf{h}_t$ (or $\mathbf{c}_t$) is the network state. The external memory $\mathbf{m}_t$ is a combination of $\mathbf{m}_{t-1}$ and the current state $\mathbf{c}_t$. If $g_{i,t} = 0$ and $g_{f,t} = 1$ for several successive time steps, the content saved in the external memory $\mathbf{m}_t$ would constitute the long-term memory of the system.

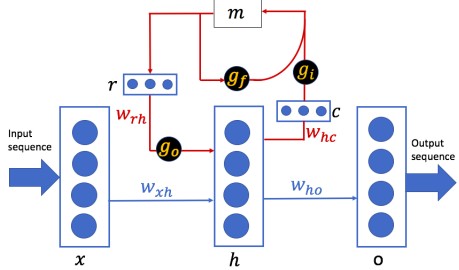

(a) Network architecture

(b) Memory visualization

Figure 3: LSTM: The blue belt, $M0$, represents the external memory. At $t_1$, memory $M00$ is generated and stored; at time $t_9$, $M00$ is updated to $M01$. The black dashed arrows represent the effect of the current state on the external memory. The state index is also the time index.

The external memory $\mathbf{m}_t$ adds greater flexibility to the state transition diagram. As shown in Fig.3(b), the current state $s_t$ (represented by hidden state $\mathbf{h}_t$) depends on either the previous state $s_{t-1}$ or the external memory $\mathbf{m}_{t-1}$ (if forget gate $g_{f,t} = 0$ and input gate $g_{i,t} = 1$, $s_t$ depends on $s_{t-1}$; if $g_{f,t} = 1$ and $g_{i,t} = 0$, $s_t$ depends on $\mathbf{m}_{t-1}$; if $0 < g_{f,t} < 1$ and $0 < g_{i,t} < 1$, $s_t$ depends on both $\mathbf{m}_{t-1}$ and $s_{t-1}$. The detailed calculations of these gates are provided in Appendix A.1). For example, $s1$ depends on the previous state $s0$, as illustrated by the blue arrows; $s7$ and $s8$ depend on the long-term memory $M00$, as illustrated by the yellow arrows; and $s9$ depends on both the previous state $s8$ and the long-term memory $M00$. The introduced external memory circumvents the compromise between the memory depth and memory resolution that is always present in the state memory in vRNN. For instance, for a 10th-order binary Markov sequence whose state dependence relationship is $s_t = f(s_{t-1}, s_{t-10})$, a vRNN has to learn a state space with $2^{10}$ states (10 states are combined into a new state); however, LSTM needs to learn a state model with only 2 states and an external memory storing the state information from 10 steps in the past. **By constructing this short path between long-term memory and the current state, LSTM performs much better than vRNN for sequences that skip intermediate values of time dependencies. This long-term memory can also be seen as the event extracted from the input time series.** The relationship of LSTM and vRNN is shown in Theorem1.

**Theorem 1.** *LSTM degrades to a vRNN if i) all three gates are constants, i.e., $g_o = 0$, $g_i = 1$ and $g_f = 0$; ii) weight $\mathbf{w}_{hc}$ and bias $\mathbf{b}_c$ are set to constants, i.e., $\mathbf{w}_{hc} = \mathbf{I}$ and $\mathbf{b}_c = \mathbf{0}$; iii) the activation function $t(x)$ is a linear activation function, i.e., $t_1(x) = x$.*

*Proof.* See Appendix B.1. □

Although LSTM is more effective than vRNN, some limitations still exist. For example, if there is no skip in the time dependence, i.e., $s_t = f(s_{t-1}, s_{t-2}, ..., s_{t-10})$, LSTM and vRNN have the same expressive power. Therefore, the statement that "LSTM is always better than vRNN" is not correct. Another drawback of LSTM is the transient storage of long-term memory. In other words, if the long-term memory is updated, the old value is erased. For example, in Fig.3(b), at time $t_9$, when M00 is updated to M01, M00 is erased. Thus, the future states no longer have access to memory M00. Therefore, this architecture is useful when the previous states do not need to be reutilized after they are updated.

### 4.3 Neural Stack

A neural stack is a neural network that uses a stack as its external memory. The stack is controlled by either a feedforward network or a vRNN. One property of a stack is that only the topmost content of the stack can be read or written. Writing to the stack is implemented via three operations: push, adding

an element to the top of the stack; pop, removing the topmost element of the stack; no-operation, keeping the stack unchanged.

A diagram of a neural stack network is shown in Fig.4(a) (here, we use the architecture in Joulin & Mikolov (2015)). Elements in the stack are updated as follows:

$$\mathbf{s}_t(i) = \begin{cases} d_t^{push}\mathbf{c} + d_t^{pop}\mathbf{s}_{t-1}(1) + d_t^{no-op}\mathbf{s}_{t-1}(0), & if\ i = 0, \\ d_t^{push}\mathbf{s}_{t-1}(i-1) + d_t^{pop}\mathbf{s}_{t-1}(i+1) + d_t^{no-op}\mathbf{s}_{t-1}(i), & otherwise, \end{cases} \quad (7)$$

$\mathbf{s}_t(i)$ is the content of the stack at time $t$ in position $i$. $\mathbf{s}_t(0)$ is the topmost content at time $t$, $\mathbf{c}$ is the candidate content to be pushed onto the stack, and $d_t^{push}$, $d_t^{pop}$ and $d_t^{no-op}$ are the push, pop and no-operation signals. All operations have to be implemented by continuous functions over a continuous domain to train the network with backpropagation through time. The calculation details of the stack contents and the corresponding operators are given in Appendix A.2. Since recurrence is introduced by stack memory, the dynamics of the model are

$$\mathbf{h}_t = g(\mathbf{w}_{xh}^T\mathbf{x}_t + \mathbf{w}_{rh}^T\mathbf{r}_t + \mathbf{b}_h), \quad (8)$$

where $\mathbf{r}_t$ is the vector read at time $t$,

$$\mathbf{r}_t = g_o\mathbf{s}_t(0). \quad (9)$$

Although the neural stack architecture looks very different from that of vRNN and LSTM, some underlying similarities exist from a memory organization perspective. Fig.4(b) shows the memory space for the neural stack. **In contrast to LSTM, a neural stack can store more than one useful event in its external memory bank.** ( In LSTM, if there is only one useful event, it can be stored as it is, but if there are multiple useful events, they have to be compressed into one compounded event and can only be accessed as a whole.) For example, at time $t_0$, $M00$ is saved in memory belt $M0$, and at time $t_2$, $M10$ is saved in

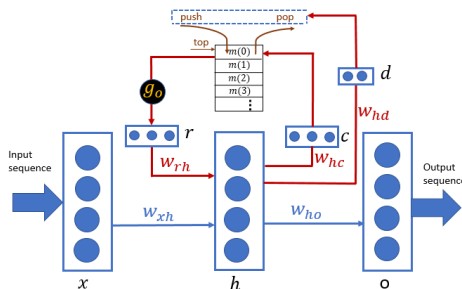

(a) Network architecture

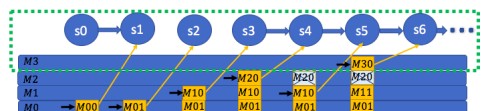

(b) Memory visualization

Figure 4: Neural stack: the network first saves state $M00$ in belt $M0$ and updates it to $M01$. At time $t_2$, instead of replacing $M01$ with a new state $M10$, a new belt $M1$ is created to save $M10$. In this way, both $M01$ and $M10$ are kept. Similarly, at time $t_5$, $M30$ is saved in another belt $M3$. At time $t_5$, the content in the stack is $M01$, $M11$, and $M30$, where $M30$ is the topmost element.

belt $M1$. The black arrow on the left of the memory content points to the top of the stack in each time step. All these contents can be addressed as needed. For example, $M10$ is used again at time $t_4$ after popping out $M20$ in belt $M2$. All the useful information of the input is retained with this external memory. In contrast to state memory, the previous content is not altered but is stored in its original or transformed form. As the past content and operations are separate, we can efficiently select the useful content from this structured memory rather than using the mixture of all previous content.

**LSTM can be seen as a special case of a neural stack as proved in Theorem 2. In a neural stack, if all the contents in the stack below the topmost element are never addressed again, a single memory belt is sufficient. In this case, the neural stack degrades to LSTM**, as shown in Fig.4(b) in the green dashed box. **The stack operators (push, pop, no-operation) in the neural stack have the same function as the input and forget gates in LSTM**: deciding how to revise the memory contents. The problem in LSTM is that the previous memory is erased after the update, which also occurs continuously with the vRNN state. Hence, both learning models have difficulties in performing some simple memorization tasks, such as reversing a sequence. However, the external memory bank in a neural stack can help to solve this problem via the online storage and extraction of more than one content.

**Theorem 2.** *The neural stack can degrade to LSTM if the pop signal is zero, i.e., $d_t^{pop} = 0$. The $d_t^{push}$ in a neural stack works as the input gate in LSTM, and $d_t^{no-op}$ in a neural stack works as the forget gate in LSTM.*

*Proof.* See Appendix B.2. □

Although the neural stack can return to previous memory, two constraints exist. First, the neural stack cannot jump to any memory position: the previous memory must be addressed and updated sequentially. For example, as shown in the second line of Fig.4(b), if we want to return to the memory in belt $M1$, we first have to pass the memory in belt $M2$. Second, the memory content is forgotten after it is popped out of the stack. For example, at time $t_4$, the memory in belt $M2$ is popped out, so in the subsequent time steps, the content in belt $M2$ can no longer be accessed and updated.

From the state transition analysis above, we can conclude that the stack neural network is the ideal choice for tasks where the previous memory must be addressed sequentially (first in last out).

### 4.4  NEURAL RAM

Dynamic neural networks with external random access memory have recently been studied. In these networks, all the contents in the memory bank can be randomly accessed. The neural Turing machine Graves et al. (2014) (NTM), whose network architecture is shown in Fig.5(a), is one example. The challenge with this network is that all the memory addresses are discrete in nature. To learn, read and write addresses by error backpropagation, the addresses have to be extended to continuous domain.

One solution to this difficulty is to read from and write to all the memory slots with different strengths. These strengths represent the probability that each slot is be read from and written to. Specifically, the reading vector at time step $t$ is

$$\mathbf{r}_t = \sum_{i=0}^{M-1} w_t^r(i)\mathbf{m}_t(i), \qquad (10)$$

where $\mathbf{m}$ is a memory bank with $M$ memory slots, $w_t^r(i)$ is the normalized reading weight for the $i$th slot at time $t$, which satisfies $\sum_i w_t^r(i) = 1, 0 \le w_t^r(i) \le 1$. In the writing process, each memory slot is updated as

$$\mathbf{c}_t = f(\mathbf{w}_{hc}^{\mathrm{T}}\mathbf{h}_t + \mathbf{b}_c), \qquad (11)$$
$$\mathbf{m}_t(i) = w_t^w(i)\mathbf{c}_t(i) + e_t(i)\mathbf{m}_{t-1}(i), \forall i \qquad (12)$$

where $w_t^r(i)$ is the writing weight, and $e_t(i)$ is the erasing weight for memory slot $i$ at time $t$. The detailed calculations of these weights are presented in Appendix A.3. The dynamics of the hidden layer are

$$\mathbf{h}_t = f(\mathbf{w}_{xh}^{\mathrm{T}}\mathbf{x}_t + \mathbf{w}_{rh}^{\mathrm{T}}\mathbf{r}_{t-1} + \mathbf{b}_h), \qquad (13)$$

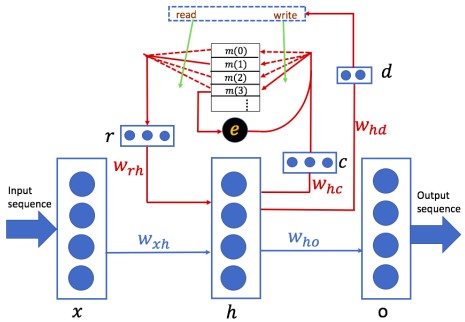

(a) Network architecture

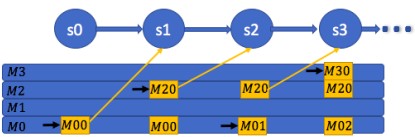

(b) Memory visualization

Figure 5: Neural RAM

Fig.5(b) shows the memory structure of neural RAM. **The RAM network can be seen as an improvement of the neural stack in the sense that all the contents in the memory bank can be read from and written to multiple times. Moreover, no requirements exist for the order of storing, updating and accessing the memory elements.** For example, in Fig.5(b), at time $t_0$, memory $M00$ is stored in belt $M0$; at time $t_1$, the system control can directly jump to belt $M2$ to store $M20$. Additionally, the reading and writing slots can be different. For example, at $t_1$, the network writes to belt $M2$ and reads from $M0$. The black arrows on the left of the contents in external memory represent the reading contents. This neural RAM network can degrade to a neural stack if the memory accessing order is restricted. Their relationship is stated in Theorem 3. Similarly, the network can degrade to LSTM if only one memory belt is used. The above analysis indicates that neural RAM is the most powerful network among the models discussed in this paper.

**Theorem 3.** *Neural RAM can degrade to a neural stack if*

*i) all the reading weights except that of the topmost memory slot are set to zero, $w_t^r(i) = 0, if\ i \ne 0$;*

*ii) only the writing weight for the topmost memory slot is learned, and all others are copied from it, $w_t^r(i) = w_t^r(0), if\ i \ne 0$;*

*iii) in the writing process, instead of learning all the contents to be written to the stack, as in Eq.(11), only the content of M0 is learned as $\mathbf{c}_t(0) = t(\mathbf{w}_{hc}^{'\mathrm{T}}\mathbf{h}_{t-1} + b_c) + \gamma\mathbf{m}_{t-1}(1)$, and all other contents are calculated as $\mathbf{c}_t(i) = \mathbf{m}_{t-1}(i-1) + \gamma\mathbf{m}_{t-1}(i+1), if\ i \ne 0$.;*

*iv) only the writing and erasing weights for the topmost element are learned, and all others are copied from the values of the topmost element, $w_t^r(i) = w_t^r(0), e_t(i) = e_t(0)$.*

*Proof.* See Appendix B.3. □

## 5 MAPPING FROM NETWORK TYPES TO TASK TYPES

From the analysis above, we know that the proposed taxonomy for the four most conventional memory architectures appears as a simple way to quantify the past information: state memory, memory of one compounded event, memory of multiple events with restricted access order, memory of multiple events with randomly access order. In this section, we will discuss how to utilize this proposed taxonomy to help practitioners select the right model.

The proposed taxonomy resembles the hierarchical organization of automata: vRNN⇔Finite state machine, neural stack⇔Deterministic pushdown automaton, neural RAM⇔Turing machine. Hence, if our task is sequence recognition or classification, the recognizable sequences for each network can be illustrated by the Chomsky hierarchy. However, these networks can perform additional sequence learning tasks, such as prediction. In this case, sequences do not need to satisfy the restricted grammars. For example, no need exists for the input sequences to always start from a start state and return to an accepted state. Hence, to ensure that our taxonomy fits these more general sequences, we have to analyze the memory requirements of the specific task first and match it with these four types of memory networks.

Four representative tasks that would fit optimally the characteristics of each memory organization are discussed here: counting, counting with interference, reversing and repeat copying. Therefore, practitioners can compare their own problem with these four tasks and get some hints to select the right model. We will analyze the memory requirements of each task individually.

**Counting**   In the counting task, the input sequences are composed of $a$'s, $b$'s and $c$'s. The output sequence is the cumulative count of the number of $a$'s. For instance, when the input sequence is $aaabcaa$, the output sequence would be 1233345. For this type of sequence, a state variable is required to remember the number of $a$'s. A state transition occurs when an $a$ is encountered. In this problem, the state space is not very large, and a first-order Markov state model provides a sufficient description. Hence, the counting task can be completed as long as the network has one feedback loop. In this paper, "task can be completed" means the output error is almost zero.

**Counting with interference**   In the counting with interference task, the input sequences are the same as those of the counting task. The goal is still to count the number of $a$'s, but if a $b$ or $c$ is encountered, the output should be $b$ or $c$. For example, if the input is $aabbaca$, the output sequence is $12bb3c4$. For this type of problem, an external memory cache is required because the hidden layer's output (internal memory) is overwritten when a $b$ or $c$ is encountered. If we want to recall the number of $a$'s, the value must be stored in external memory for future use, and an input gate must be used to protect the external memory when inputting $b$ and $c$ ($g_i = 1$ when the input is $a$, and $g_i = 0$ when the input is $b$ or $c$). Thus, LSTM, neural stack and neural RAM are capable of solving this problem. However, in a vRNN, since the only memory is the state memory and the output is forced to be a function of this state memory, the interference of $b$ and $c$ prevent a vRNN from accomplishing this task.

**Reversing**   The third task is sequence reversing. For example, if the input sequence is $abacde\delta xxxxxxx$, the output sequence should be $xxxxxxxedcaba$. $\delta$ is the delimiter symbol, and $x$ represents any symbol. When $\delta$ is encountered in the input sequence, regardless of the subsequent symbols, the output is the input symbols before $\delta$ in reverse order. For this task, all the useful past information should be stored and then retrieved in reverse order. Hence, the memory must have the ability to store more than one element, and the reading order is related to the writing order. Since vRNN does not have a memory bank and LSTM's memory is forgotten after being updated, these two networks fail in this task. By contrast, the neural stack and neural RAM can both store more than one content and can solve this task, which satisfy the "first in last out" principle.

**Repeat copying**   The most challenging task is repeat copying, where the output sequence is the input sequence repeated several times. For example, if the input sequence is $adbc\varepsilon xxxxxxxxxxxxxxx$, the output should be $xxxxxxxadbcadbcadbc$. That is, when the repeating number symbol $\varepsilon$ is

encountered, the output is the previous input sequence repeated $\varepsilon$ times. For this type of task, more than one past content must be stored and retrieved multiple times, in this case, 3 times. Since all the saved information in a neural stack is forgotten after being popped out, a neural stack cannot perform this task; thus, only a neural RAM can complete this task.

This classification of tasks is meaningful since it can guide users in the right direction. If we select the wrong type of network, an error and/or speed penalty will occur regardless of how we adjust the hyperparameters. As shown in our experiment, for sequence reversing (i.e., the third type of task), neural stack and neural RAM with 6 hidden neurons converge to near zero error, but for vRNN and LSTM, even if we set the number of hidden neurons to 1000, the output will always fluctuate around a nonzero value.

## 6 EXPERIMENTS

We test the performance of the four networks on the synthetic tasks described in last section to illustrate the impact of different memory organizations. We also visualize how each network encodes information to solve a problem in AppendixD. Then, we use two natural language processing applications to elaborate how to employ the knowledge gained from the different characteristics of the memory structures to help users to select the appropriate type of network. The details of parameters settings are given in AppendixC.

### 6.1 SYNTHETIC TASKS

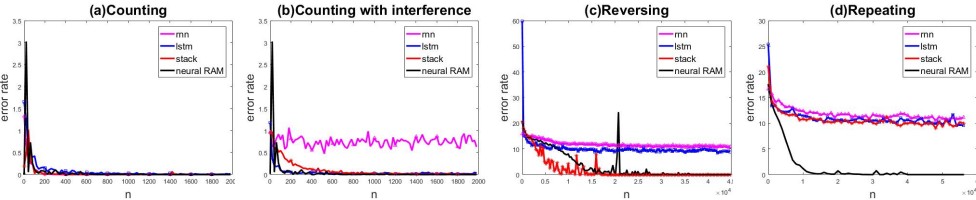

Figure 6: Learning curves for four synthetic tasks

The learning curves for the four tasks when using different networks are shown in Fig.6. The performance is measured in terms of MSE for first two tasks and output entropy for the last two tasks. We use the same number of units in all the architectures to ensure a fair comparison. The results indicate that for counting, all four networks can achieve an almost zero error; for counting with interference, all the networks except vRNN can complete the task; for sequence reversing, neural stack and neural RAM are suitable networks; and for repeat copying, neural RAM is the only network that can solve the problem. We also apply different parameter settings, for instance, varying the number of hidden units from 5 to 1000. The performances are the same as shown in Fig.6, except a different nonzero error value is observed when the network is not capable of accomplishing the task.

### 6.2 NATURAL LANGUAGE PROCESSING

Synthetic problems could be designed to exemplify the expressive power of the different memory networks. For real-world problems, this task is more complex because the memory requirements of the problem may be difficult to determine and the problem may be a blend of classes. In this case, all the networks may be able to solve the problem to a certain extent. Hence, in this section, we illustrate how to select the minimum network resources to accomplish the task with relatively good performance.

**Sentiment analysis** The first experiment is sentiment analysis to infer the emotional tone of a text as negative or positive. The sentiment of a text is mainly based on the occurrence and reoccurrence of some discriminating words. Specifically, if there are a lot of positive words, such as 'good, love' in the text, the sentiment of the text is always positive, and vice versa. Hence, external memory whose value is affected by these discriminating words is useful to judge the emotional tone of text. Thus LSTM, neural stack and neural RAM should perform better than vRNN.

Table 2: Average error for movie review

|  | vRNN | LSTM | neural Stack | neural RAM |
|---|---|---|---|---|
| error rate | 31±5 | **19±2.5** | 23±10 | 20±9 |

Table 3: Average error for three tasks from bAbI tasks

| Task | vRNN | LSTM | neural Stack | neural RAM |
|---|---|---|---|---|
| task 1 | 52±1.5 | 28.4±1.5 | 41±2.0 | **9.0±12.6** |
| task 2 | 79±2.5 | 56.0±1.5 | 75±6 | **39.2±20.5** |
| task 3 | 85±2.5 | 51.3±1.4 | 78±6.4 | **39.6±16.4** |

Since the goal is to classify the emotional tone as either 1 or 0, the specific contents of the text are not important and thus do not need to be all stored. For example, as long as it is a positive word, whether it is 'love''good' or 'like' is not very important. Hence, multi-slot memory which can store multiple contents does not have more advantages than the single-slot memory.

We test the network performance on the IMDb movie review dataset Pennington et al. (2014). The results are presented in Table 2, which shows that vRNN performs worst. LSTM, neural stack, and neural RAM have similar performances; thus, our analysis is verified.

**Question answering** The second experiment investigates the performance of these four networks on three question answering tasks from the bAbI datasetWeston et al. (2015). The target is to answer a question after reading a short story. For example, for the story "Mary got the milk there. John moved to the bedroom. Sandra went back to the kitchen. Mary travelled to the hallway.", and the question "Where is the milk?", the expected answer is "hallway". For this problem, to give the correct answer, the network must memorize the facts that Mary got the milk and traveled to the hallway. Additionally, since the does not know the question before reading the story, all the potentially useful facts must be stored. Thus, an external memory bank in which any content can be visited is useful. According to our memory capability analysis, the neural RAM should perform the best in this scenario. The results in Table 3 confirm that neural RAM achieves the best performance.

From the results we can see that, neural stack's performance is worse than LSTM. We know that both LSTM and neural stack are not suitable for this task. If we apply LSTM or neural stack to the problem, they will try their best to find their approximate solutions. If we apply LSTM to this task, LSTM would compress all the useful information in its external memory, although the right answer is mixed with other information, the output can at least get some information from it. However, if we apply neural stack to this task, the push signal sometimes is very large ($d^{push} \approx 1, d^{pop} \approx 0, d^{no-op} \approx 0$), which means that the right answer will then be pushed down in the stack which cannot be accessed when needed. Hence, we think although the neural stack attempts to find the right answer, it is more likely to be stuck in local points due to its complicated operation. But we have to be aware that it is not due to the memory limitation of stack itself, a better implementation of the three operators may help improve neural's stack performance for this kind of problem.

## 7    CONCLUSION

In this paper, we analyze the memory structures of several recurrent networks and propose a taxonomy. We use four synthetic tasks and two natural language processing problems to illustrate utility of the taxonomy. Although we show differences in performance in the experiments, it is too early to say that we presented all the tools necessary to select the parsimonious memory architecture for a given application. Because the user has to analyze the requirements of the application, which may not be trivial, additionally work is needed to create guidelines to assist practitioners.

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

## A  NETWORK ARCHITECTURE COMPONENTS

### A.1  LSTM

The three gates are in LSTM are calculated as follows:

$$g_{i,t} = s(\mathbf{w}_{hg_i}^{\mathrm{T}}\mathbf{h}_{t-1} + \mathbf{w}_{xg_i}^{\mathrm{T}}\mathbf{x}_t + \mathbf{b}_{g_i}), \tag{14}$$

$$g_{f,t} = s(\mathbf{w}_{hg_f}^{\mathrm{T}}\mathbf{h}_{t-1} + \mathbf{w}_{xg_f}^{\mathrm{T}}\mathbf{x}_t + \mathbf{b}_{g_f}), \tag{15}$$

$$g_{o,t} = s(\mathbf{w}_{hg_o}^{\mathrm{T}}\mathbf{h}_{t-1} + \mathbf{w}_{xg_o}^{\mathrm{T}}\mathbf{x}_t + \mathbf{b}_{g_o}). \tag{16}$$

where $\mathbf{w}_{hg_i}, \mathbf{w}_{hg_f}, \mathbf{w}_{hg_o}$ are $K_h \times 1$ weights , $\mathbf{w}_{hg_i}, \mathbf{w}_{hg_f}, \mathbf{w}_{hg_o}$ are $K_i \times 1$ weights and $b_{g_i}, b_{g_i}$ and $b_{g_i}$ are bias. These three gates give flexibility to operate on memories.

### A.2  NEURAL STACK

Neural network interacts with the stack memory by $d_t^{push}, d_t^{pop}, d_t^{no-op}, \mathbf{c}_t$ and $\mathbf{r}_t$. According to Sun (1993) the domain of the operations is relaxed to any real value in $[0, 1]$. This extension adds a continuous amplitude dimension to the operations. For example, if the push signal $d_{push} = 1$, the current vector will be pushed into the stack as it is, if $d_{push} = 0.8$, the current vector is first multiplied by 0.8 and then pushed into the stack.

$d_t^{push}$, $d_t^{pop}$, $d_t^{no-op}$ and $\mathbf{c}_t$ are decided by the hidden layer outputs and the corresponding weights,

$$\mathbf{d} = [d_t^{push}, d_t^{pop}, d_t^{no-op}]^{\mathrm{T}} = s(\mathbf{w}_{hd}^{\mathrm{T}}\mathbf{h}_t + \mathbf{b}_{op}),$$

where $\mathbf{w}_{hd}$ is the $K_h \times 3$ weights and $\mathbf{b}_{op}$ is the $3 \times 1$ bias.

$$\mathbf{c}_t = g(\mathbf{w}_{hc}^{\mathrm{T}}\mathbf{h}_t + \mathbf{b}_c),$$

where $\mathbf{w}_{hc}$ is the $K_h \times N$ weights and $\mathbf{b}_{op}$ is the $N \times 1$ bias. Here we assume all the elements saved in the stack are $N \times 1$ vectors.

## A.3 NEURAL RAM

In the neural RAM, the read weighting $w_t^r(i)$ is learned as,

$$\boldsymbol{w}_t^r = f(\mathbf{w}_{ha}^{\mathrm{T}}\mathbf{h}_{t-1} + \mathbf{b}_r), \tag{17}$$

$\mathbf{w}_{ha}$ is the $K_h \times M$ weight and $\mathbf{b}_a$ is $M \times 1$ bias, $\boldsymbol{w}_t^r = [w_t^r(0), w^r(1), ..., w^r(M-1)]^{\mathrm{T}}$ The nonlinear activation function $f$ is usually set as softmax function. The write weighting is learned as,

$$\boldsymbol{w}_t^w = s(\mathbf{w}_{hb}^{\mathrm{T}}\mathbf{h}_{t-1} + \mathbf{w}_{xb}^{\mathrm{T}}\mathbf{x}_t + \mathbf{b}_w). \tag{18}$$

where $\mathbf{w}_{hb}$ is $K_h \times M$ weight, $\mathbf{w}_{xb}$ are $K_i \times M$ weight, $\mathbf{b}_b$ is $M \times 1$ bias.

And the erase weighting is learned as,

$$\mathbf{e}_t = s(\mathbf{w}_{he}^{\mathrm{T}}\mathbf{h}_{t-1} + \mathbf{w}_{xe}^{\mathrm{T}}\mathbf{x}_t + \mathbf{b}_e), \tag{19}$$

where $\mathbf{e}_t = [e_t(1), e_t(2), ..., e_t(M)]^{\mathrm{T}}$, $\mathbf{w}_{he}$ is $K_h \times M$ weights, $\mathbf{w}_{xe}$ is $K_i \times M$ weight, and $\mathbf{b}_e$ are $M \times 1$ bias. In practice, instead of learning the read and write head from scratch, some methods were proposed to simplify the learning process. For example in Neural Turing machine Graves et al. (2014), $e_t(i)$ is coupled with write weight $w_t^w(i)$, $e_t(i) = 1 - w_t^w(i)$. And reading weight $\boldsymbol{w}_t^r$ and writing weight $\boldsymbol{w}_t^w$ are obtained by content-addressing and location-addressing mechanisms. The content-addressing mechanism gives the weights $w_t^r(i)$ (or $w_t^w(i)$) by checking the similarity of the key $\mathbf{d}$ with all the contents in the memory, the normalized version is,

$$w_t^r(i) = \frac{exp(\alpha K[\mathbf{d}, \mathbf{m}_t(i)])}{\sum_j(exp(\alpha K[\mathbf{d}, \mathbf{m}_t(j)]))},$$

where $\alpha$ is the parameter to control the precision of the focus, $K$ is a similarity measure. Then, the weights will be further adapted by the location-addressing mechanism. For example, the weights obtained by content addressing can firstly blend with the previous weight and then shifted for several steps,

$$w_t^r(i) = g_t w_{t-1}^r(i) + (1 - g_t)w_t^r(i), \tag{20}$$
$$w_t^r(i) = w_t^r([i - n]_M). \tag{21}$$

$g_t$ is the gate to balance the previous weight and current weight, $n$ is the shifting steps, $[i - n]_M$ means the circular shift for $M$ entities. Since the shifting operation is not differentiable, the method in Graves et al. (2014) should be utilized as an approximation.

Another example is Graves et al. (2016) which improves the performance even more. To be specific, for reading, a matrix to remember the order of memory locations they are written to can be introduced. With this matrix, the read weight is a combination of the content-lookup and the iterations through the memory location in the order they are written to. And for writing, a usage vector is introduced, which guides the network to write more likely to the unused memory. With this modification, the neural RAM gets flexibility similar to working memory of human cognition which makes it more suitable to intelligent prediction. With these modifications, the training time for the neural RAM is also reduced.

## B THEOREM PROOFS

### B.1 PROOF OF THEOREM 1

Compared to vRNN, LSTM introduces an external memory and the gate operation mechanism. So if we set the output gate $g_o = 0$, input gate $g_i = 1$ and the forget gate $g_f = 0$ instead of learning from

the sequences, the dynamics of LSTM is degraded to vRNN as follows,

$$\mathbf{h}_t = t(\mathbf{w}_{xh}^{\mathrm{T}}\mathbf{x}_t + \mathbf{w}_{rh}^{\mathrm{T}}g_o\mathbf{r}_t + \mathbf{b}_h) \tag{22}$$

$$= t[\mathbf{w}_{xh}^{\mathrm{T}}\mathbf{x}_t + \mathbf{w}_{rh}^{\mathrm{T}}\mathbf{r}_r + \mathbf{b}_h] \tag{23}$$

$$= t[\mathbf{w}_{xh}^{\mathrm{T}}\mathbf{x}_t + \mathbf{w}_{rh}^{\mathrm{T}}\mathbf{m}_t + \mathbf{b}_h]$$

$$= t[\mathbf{w}_{xh}^{\mathrm{T}}\mathbf{x}_t + \mathbf{w}_{rh}^{\mathrm{T}}(g_i\mathbf{c}_t + g_f\mathbf{m}_{t-1}) + \mathbf{b}_h]$$

$$= t(\mathbf{w}_{xh}^{\mathrm{T}}\mathbf{x}_t + \mathbf{w}_{rh}^{\mathrm{T}}\mathbf{c}_t + \mathbf{b}_h) \tag{24}$$

$$= t[\mathbf{w}_{xh}^{\mathrm{T}}\mathbf{x}_t + \mathbf{w}_{rh}^{\mathrm{T}}t_1(\mathbf{w}_{hc}^{\mathrm{T}}\mathbf{h}_{t-1} + \mathbf{b}_c) + \mathbf{b}_h] \tag{25}$$

$$= t(\mathbf{w}_{xh}^{\mathrm{T}}\mathbf{x}_t + \mathbf{w}_{rh}^{\mathrm{T}}\mathbf{h}_{t-1} + \mathbf{b}_h) \tag{26}$$

Here (23) is due to $g_o = 0$, (24) is due to $g_i = 1$ and $g_f = 0$, (26) is because the weight $\mathbf{w}_{hc}$ and bias $\mathbf{b}_c$ are set as constants and the activation function $t(x)$ is set as linear activation function,

$$\mathbf{w}_{hc} = \mathbf{I},$$

$$\mathbf{b}_c = \mathbf{0},$$

$$t_1(x) = x.$$

Since Eq.(22) is the dynamic of LSTM and Eq.(26) is the dynamic of vRNN, the argument that vRNN is a special case of LSTM is proved.

## B.2 PROOF OF THEOREM 2

The dynamic of LSTM is,

$$\mathbf{h}_t = f(\mathbf{w}_{xh}^{\mathrm{T}}\mathbf{x}_t + \mathbf{w}_{rh}^{\mathrm{T}}g_{o,t}\mathbf{r}_t + \mathbf{b}_h), \tag{27}$$

And the dynamic of neural stack is,

$$\mathbf{h}_t = g(\mathbf{w}_{xh}^{\mathrm{T}}\mathbf{x}_t + \mathbf{w}_{rh}^{\mathrm{T}}\mathbf{r}_t + \mathbf{b}_h), \tag{28}$$

According to Eq.(27) and Eq.(28), the dynamics of the neural stack have similar form as LSTM except for the reading vector, i.e., the reading vector is $\mathbf{r}_t = g_o\mathbf{s}_t(0)$. If we set the pop signal as zero, $d_t^{pop} = 0$, and no operation on the stack contents except for the topmost elements is available, then

$$\mathbf{s}_t(0) = d_t^{push}\mathbf{c} + d_t^{no-op}\mathbf{s}_{t-1}(0),$$

$$\mathbf{s}_t(i) = 0, \ if \ i \neq 0.$$

Since the $d_t^{push}$, $d_t^{no-op}$ are calculated in the same way the input gate $g_{i,t}$ and forget gate $g_{f,t}$ are calculated in LSTM as shown in Eq.(14) to Eq.(15), the read vector would be,

$$\mathbf{r}_t = g_o\mathbf{s}_t(0)$$

$$= g_o(d_t^{push}\mathbf{c} + d_t^{no-op}\mathbf{s}_{t-1}(0))$$

In this manner, this is exactly how the LSTM organizes its memory,

$$\mathbf{r}_t = g_o(g_{i,t}\mathbf{c} + g_{f,t}\mathbf{s}_{t-1}(0))$$

$d_t^{push}$ can be seen as the input gate and $d_t^{no-op}$ can be seen as the forget gate. Hence, it is proved that LSTM can be seen as a special case of neural stack.

## B.3 PROOF OF THEOREM 3

Neural RAM is more powerful than neural stack because it has access to all the contents in the memory bank. If we restrict the read and write vector, neural RAM is degraded to neural stack. To be specific, for the read head $\boldsymbol{w}_t^r$, all the read weights except the topmost are set to zeros,

$$w_t^r(i) = \begin{cases} 0, & if \ i \neq 0 \\ t(\mathbf{w}_{ha}'^{\mathrm{T}}\mathbf{h}_{t-1} + b_a), & if \ i = 0 \end{cases}, \tag{29}$$

here $\mathbf{w}'_{ha}$ is $K_h \times 1$ vector and $b_a$ is the scalar. Eq.(29) is a special case of Eq.(17).

$$\mathbf{c}_t(0) = t(\mathbf{w}_{hc}^{'\mathrm{T}}\mathbf{h}_{t-1} + b_c) + \gamma\mathbf{m}_{t-1}(1), \tag{30}$$

all others are calculated as,

$$\mathbf{c}_t(i) = \mathbf{m}_{t-1}(i - 1) + \gamma\mathbf{m}_{t-1}(i + 1), \; if \; i \neq 0. \tag{31}$$

And in the writing process, Eq.(30), Eq.(31) can be seen as a special case of Eq.(11) because $\mathbf{h}_{t-1}$ in (11) is a function of $\mathbf{m}_{t-1}$. Substitute (30)(31) into the memory update equation of neural RAM (12), we get,

$$
\begin{aligned}
\mathbf{m}_t(i) &= w_t^w(i)\mathbf{c}_t(i) + e_t(i)\mathbf{m}_{t-1}(i) \\
&= \begin{cases}
w_t^w(i)t(\mathbf{w}_{hc}^{'\mathrm{T}}\mathbf{h}_{t-1} + b_c) + \\
\gamma w_t^w(i)\mathbf{m}_{t-1}(1) + e_t(i)\mathbf{m}_{t-1}(i), & if\, i = 0 \\
w_t^w(i)\mathbf{m}_{t-1}(i - 1) + \\
\gamma w_t^w(i)\mathbf{m}_{t-1}(i + 1)\mathbf{c}_t(i) + e_t(i)\mathbf{m}_{t-1}(i), & otherwise
\end{cases} \\
&= \begin{cases}
w_t^w(i)t(\mathbf{w}_{hc}^{'\mathrm{T}}\mathbf{h}_{t-1} + b_c) + \\
\gamma w_t^w(i)\mathbf{m}_{t-1}(1) + e_t(i)\mathbf{m}_{t-1}(i), & if\, i = 0 \\
w_t^w(i)\mathbf{m}_{t-1}(i - 1) + \\
\gamma w_t^w(i)\mathbf{m}_{t-1}(i + 1)\mathbf{c}_t(i) + e_t(i)\mathbf{m}_{t-1}(i), & otherwise
\end{cases}
\end{aligned}
\tag{32}
$$

Finally since Eq.(3) and Eq.(3) can be seen as a special case of Eq.(18) and Eq.(19), the neural stack can be treated as a special case of neural RAM. Compared the memory writing operation of neural stack (7) and neural RAM (32), we can see that, $w_t^w(0)$, $\gamma w_t^w(0)$ and $e_t(0)$ works as the push, pop and no-operate operations respectively. Compared the memory reading operation of neural stack and neural RAM, we can see that we can see that, $w_t^r(0)$ in neural RAM (29) works as the output gate in neural stack (9).

## C  Experiment parameters setting

### C.1  Counting and counting with interference

In the experiment, the activation function is Relu in vRNN. In LSTM, the external memory's content is initialized as zero. In the neural stack, the push, pop and no-op operations are initialized as random numbers with mean 0 and variance 1. At first, there is only one content in the stack which is initialized as zero. The depth of the stack can increase to any number as required. In neural RAM, memory depth is set as $M = 3$. In LSTM, neural stack and neural RAM, memory width is set as $N = 3$, the nonlinear activation functions for all the gates are sigmoid functions and others are tanh. The number of input neurons, hidden neurons and output neurons are 3. All the weights are initialized as random numbers with mean 0 and variance 1, all the bias are initialized as 0.1. For counting task, the model is trained with the synthetic sequences up to length 20. When the input is $a$, the first elements in the output vector would add one, otherwise, the output vector is unchanged. After encoding, the input and output vectors are,

| time step | input sequence | output sequence |
|-----------|----------------|-----------------|
| 1 | [1 0 0] | [1 0 0] |
| 2 | [1 0 0] | [2 0 0] |
| 3 | [0 1 0] | [2 0 0] |
| 4 | [0 1 0] | [2 0 0] |
| 5 | [1 0 0] | [3 0 0] |
| 6 | [0 0 1] | [3 0 0] |
| 7 | [1 0 0] | [4 0 0] |

For counting with interference task, after encoding, the input and output vectors are,

| time step | input sequence | output sequence |
|:---:|:---:|:---:|
| 1 | [1 0 0] | [1 0 0] |
| 2 | [1 0 0] | [2 0 0] |
| 3 | [0 1 0] | [0 1 0] |
| 4 | [0 1 0] | [0 1 0] |
| 5 | [1 0 0] | [3 0 0] |
| 6 | [0 0 1] | [0 0 1] |
| 7 | [1 0 0] | [4 0 0] |

## C.2  REVERSING AND REPEAT COPYING

Some network settings are different from the first two experiments. In vRNN, the activation function in the hidden layer is sigmoid function since we use entropy instead of mean square error as the cost function. In neural RAM, the word size and memory depth are set as 16. The length of read and write vectors are also set as 16. The number of input neurons, hidden neurons and output neurons are 6, 64, 6. The model is trained with sequences up to length 20. In repeat copying experiment, the training sequences are composed of a starting symbol $\epsilon$, some symbols in set $\{a, b, c, d, e\}$ followed by a repeating number symbol $\delta$ and some random symbols. $\epsilon$, $a$, $b$, $c$, $d$, $e$ are one-hot encoded with on value 1 and off value 0; $\delta$ is encoded with on value $n$ and off value 0, $n$ is the repeating number.

## C.3  SENTIMENT ANALYSIS

In our experiments, the number of input neurons, hidden neurons and output neurons are 50, 64, 2 for all the four network architectures. After encoding all the words into vectors, they are fed into the network one by one. Here we use a pretrained model: GloVe Pennington et al. (2014) to create our word vector. The matrix contains 400,000 word vectors, each with a dimensionality of 50. The matrix is created in a way that words having similar definitions or context reside in the relatively same position in the vector space. The decision of the tone of the paragraph will be made at the end of the paragraph. The output is [1, 0] for the positive text and [0, 1] for the negative text. The dataset adopted here is the lmdb movie review dataMaas et al. (2011) which has 12500 positive reviews and 12500 negative reviews. Here we use 11500 reviews for training and 1000 data for testing. In our experiments, the nonlinear activation functions for all the gates are sigmoid. The activation functions at the output layer are sigmoid and others are tanh. In neural RAM, the word size and memory depth are set as 64. The number of read and write head are 4 and 1. The results are average of 20 runs with random initializations.

## C.4  QUESTION ANSWERING

For each task, we use the 10,000 questions to train and report the error rates on the test set. The experimental settings for LSTM and neural RAM are the same as Graves et al. (2016) and the results for these two networks are from Graves et al. (2016). In vRNN and neural stack, the nonlinear activation functions for all the gates are sigmoid. The activation functions at the output layer are sigmoid and others are tanh. The number of input neurons, hidden neurons and output neurons are 150, 64, 150. The memory width for neural stack is 64.

## D  MEMORY WORKING PROCESS VISUALIZATION FOR THE SYNTHETIC TASKS

## D.1  COUNTING

Fig.7 to Fig.10 show the details of the memory contents after the models are well-trained. They are tested on a input sequence $bbacacbababababcc$. The little checks on the left side of each image mark the memories which affect the output results. In Fig.7, when $a$ is received, the first element of the hidden layer is increased by 1. In Fig.8, when $a$ is received, the first element of $M0$ is decreased by around 0.15. As long as there is at least one element ($h0$ in vRNN and the first element of $M0$ in LSTM) in the memory learning the pattern, after multiplying with the weight vector, the output of the network can give the expected values. Fig.9 shows how the neural stack uses its memory. Although the neural stack has the potential to use unbounded number of stack contents, it only uses the topmost content ($M0$) here, i.e. the push and no-operation cooperate to learn the pattern. The contents pushed

into $M1$ to $M5$ are never used again. Fig.10 shows the memory contents of the neural RAM and the corresponding operations of it. From Fig.10(a), we can see that $M0$, $M1$ and $M2$ cooperate to learn the pattern. When $a$ is received, the values in $M0$, $M1$ and $M2$ are increased. The read weights for $M0$, $M1$ and $M2$ are all around 0.3 as shown in Fig.10(b). From this experiment we can see that $M1$ and $M2$ are redundant here.

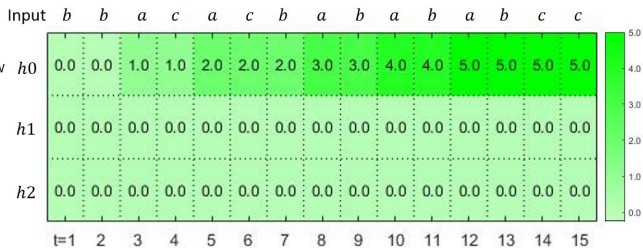

Figure 7: Task1: vRNN: Internal memory content

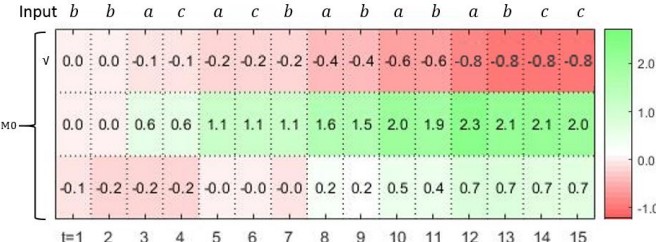

Figure 8: Task1: LSTM: External memory content

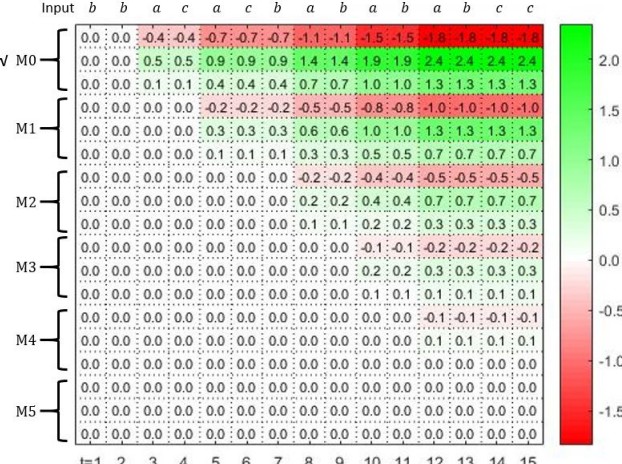

Figure 9: Task1: Neural stack: stack content

## D.2    COUNTING WITH INTERFERENCE

The experiment settings are same as "counting" task. Fig.11 to Fig.13 shows the memory usage of LSTM, neural stack and neural RAM. Fig.11 shows that when $a$ is received, the third element of the memory content would increase by around 0.2. Fig.12 and Fig.13 also show the similar incremental patterns of neural stack and neural RAM. An notable difference between Fig.11-Fig.12 and Fig.8-Fig.9 is the usage of the memory. When dealing with counting task, the output gates are always 1, however, when dealing with counting with interference task, the output gates are 0 when inputting $b$ and $c$, this helps to cut off the interference from the memory. Similarly, the read vector

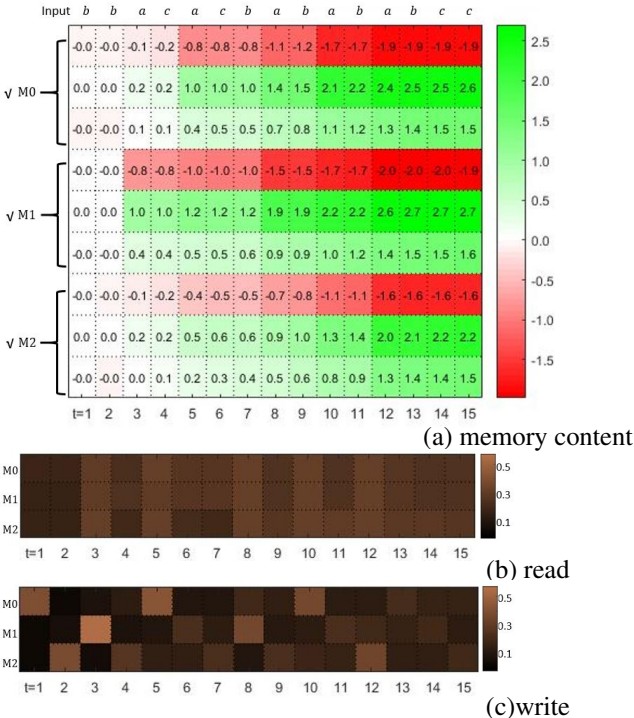

(a) memory content

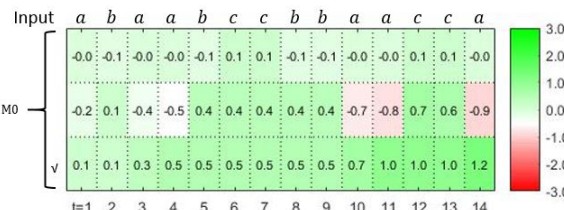

(b) read

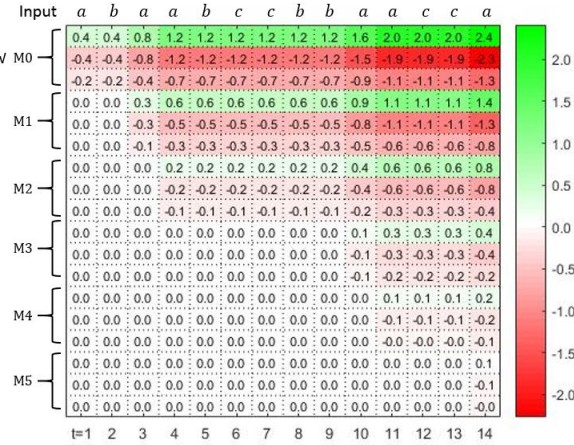

(c)write

Figure 10: Task1: Neural RAM: Memory bank content and corresponding read and write operation

Figure 11: Task2: LSTM: External memory content

Figure 12: Task2: Neural stack: stack content

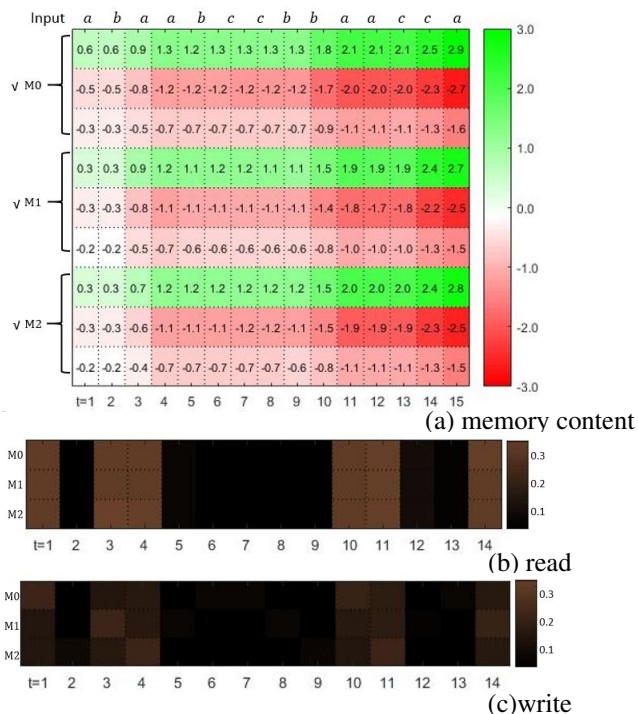
(a) memory content

(b) read

(c)write

Figure 13: Task2: Neural RAM: Memory bank content and corresponding read and write operation

are always around [0.3, 0.3, 0.3] in Fig.10, however, in Fig.13, the read vector's elements are almost zeros when encountering $b$ and $c$. The read vector here works as the output gate in LSTM and neural stack. It also shows why neural RAM does not need an output gate.

### D.3 REVERSING

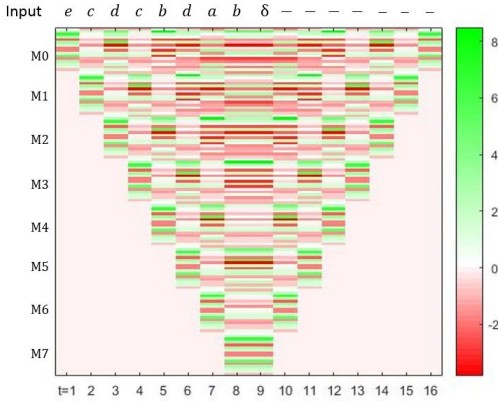

Figure 14: Task3: Neural stack: stack content

Fig.14 shows how neural stack utilizes its stack memory to solve this problem. Since each memory bank's word size is 16, here we only use colors instead of the specific numbers to show the values of contents in memory. Different from the first two tasks, the function of the stack is finally exploited. In the first half sequence, the input symbols are encoded as 16-elements vectors and pushed into the stack. In the second half of the sequence, the contents in the stack are popped out sequentially. It should be noticed that as long as the contents are popped out, they can not be revisited anymore. Different from neural stack, the contents in neural stack are never wiped as shown in Fig.15. The contents in the memory bank are only wiped if they are useless in the future or they have to be wiped

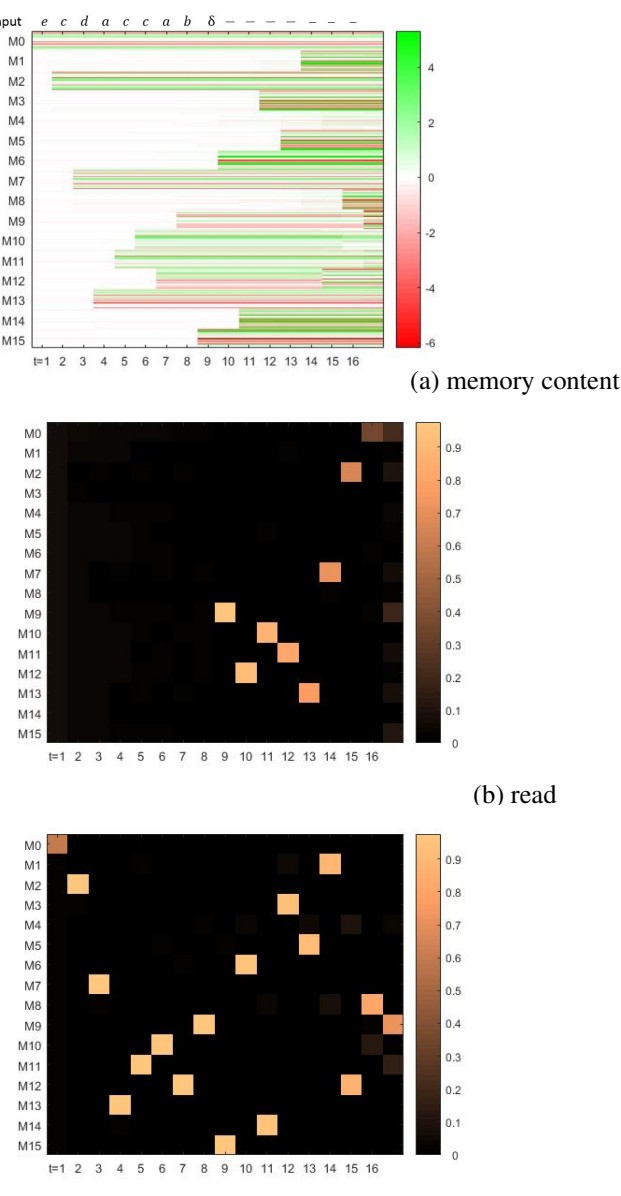

(a) memory content

(b) read

(c) write

Figure 15: Task3: Neural RAM: Memory bank content and corresponding read and write operation

to make space for new stuffs. Another feature of the memory bank for neural RAM is the memory banks are not used in order such as M0, M1, M2...In this example, the memory banks are used in the order M0, M2, M7, M13.... But as long as the network knows the writing order, the task can be accomplished. Fig15(b)(c) shows the reading and writing weights, we can see that the second half of the reading weights is the mirror of the first half of the sequence of the writing weights, which means the network learns to reverse.

## D.4 REPEAT COPYING

Since neural RAM is the only network that can handle this problem, Fig.16 shows how the neural RAM solves this problem. From the writing weights we can see that, the starting symbol is saved in M0, and symbols needed to be repeated are save in M2, M4, M6, M9. After t=4, the network would read from M2/M5, M4,M6,M9. At the beginning of every loop, the network reads from both M2 and M5 probably because the repeating time symbol $\delta$ is saved in M5. The value in M5 can tell the

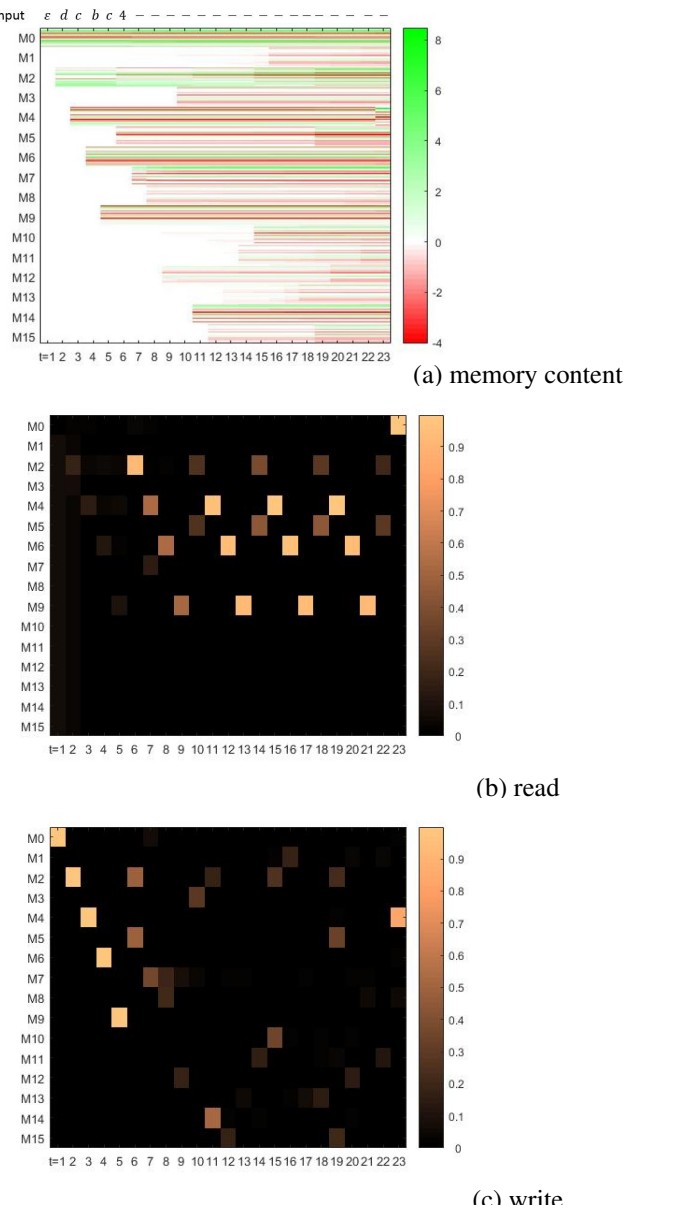

(a) memory content

(b) read

(c) write

Figure 16: Task4: Neural RAM: Memory bank content and corresponding read and write operation

network whether to continue repeating or not. We can see from Fig.16(b), at time t=22, after reading from M2 and M5, the network stops reading from M4 to M9 and turns to M0.

