# OpenReview forum: "Analysis of Memory Organization for Dynamic Neural Networks"
_ICLR.cc/2019/Conference_

### Official Review · AnonReviewer2 · 2018-10-27
**Taxonomy is not illuminating**

**Rating:** 3
**Confidence:** 5

**Review:**

The authors propose a review-style overview of memory systems within neural networks, from simple RNNs to stack-based memory architectures and NTM / MemNet-style architectures. They propose some reductions to imply how one model can be used (or modify) to simulate another. They then make predictions about which type of models should be best on different types of tasks.

Unfortunately I did not find the paper particularly well written and the taxonomy was not illuminating for me. I actually felt, in the endeavor of creating a simple taxonomy the authors have created confusing simplifications, e.g.

"LSTM: state memory and memory of a single external event"

to me is mis-leading as we know an LSTM can compress many external events into its hidden units. Furthermore the taxonomy did not provide me with any new insights or display a prediction that was actually clairvoyant. I.e. it was clear from the outset that a memory network (say) will be much better at bAbI than a stack-augmented neural network. It would be more interesting to me, for example, if the paper could thus formalize why NTMs & DNCs (say) do not outperform LSTMs at language modeling, for example. I found the reductions somewhat shady, e.g. the RAM simulation of a stack is possible, however the model could only learn the proposed reduction if the number of write heads was equal to the number of memory slots --- or unless it had O(N) thinking steps per time step, where N is the number of memory slots, so it's not a very realistic reduction. You would never see a memory network, for example, simulating a stack due to the fixed write-one-slot-per-timestep interface.

Nit: I'm not sure the authors should be saying they 'developed' four synthetic tasks, when many of these tasks have previously been proposed and published (counting, copy, reverse copy).

---

> ### Author Response · Authors · 2018-11-22
> **Reply to AnonReviewer2**
>
> The intent of our paper was to analyze the type of memory utilized by different architectures to solve sequence learning problems. This is not an easy issue because ‘memory’ is a very abstract concept and the specific memory requirements for a specific task are implicit, which means that quantitatively conceptualizing and analyzing memory is a very hard problem. Cognitive scientists have defined many different types of memory, which shows the richness of the topic, and there are only a few engineering quantifiers of memory such as memory depth and memory resolution, but they are not enough for the ever-growing applications of machine learning. Hence memory quantification is lacking in the current machine learning literature and it is our main contribution.  The proposed taxonomy for the four most conventional memory architectures appears as a simple way to quantify the capabilities of extracting past information of each class.
>
> Our goal of providing methodologies for the practitioner relegated to a second objective of the paper. It is clear from your questions that our writing was not successful, and we have modified the writing in the final submission to make this point more explicit. As far as we know, our paper addresses for the first time how to exploit the knowledge gained from the different characteristics of the memory within the taxonomy to help users select the type of memory network for an application. However, we agree that the analysis is not complete yet because on the one hand users have to analyze task’s memory requirements by themselves which is not trivial and on the other hand, the algorithm accuracy is also affected by the size and specific network structure even within a given class of models. But we firmly believe that classifying memory architectures into these four classes and linking the architecture of the learning machine to its descriptive power, as we did in this paper, is a fundamental first step. At least, in this respect, we think this paper is important to the machine learning community.
>
>
>
> Q1) I actually felt, in the endeavor of creating a simple taxonomy the authors have created confusing simplifications, e.g. "LSTM: state memory and memory of a single external event" to me is mis-leading as we know an LSTM can compress many external events into its hidden units.
>
> We agree with your statement and we are sorry for the misleading “single event”. By “single event” we mean that if there is only one useful event, it can be stored as it is, but if there are multiple useful events, they have to be compressed into one compounded event and can only be accessed as a whole. This has been clarified in the text.
>
> Q2) It would be more interesting to me, for example, if the paper could thus formalize why NTMs & DNCs (say) do not outperform LSTMs at language modeling, for example.
>
> Please see our reply to Q4) of the second reviewer.
>
> Q3) I found the reductions somewhat shady, e.g. the RAM simulation of a stack is possible, however the model could only learn the proposed reduction if the number of write heads was equal to the number of memory slots --- or unless it had O(N) thinking steps per time step, where N is the number of memory slots, so it's not a very realistic reduction. You would never see a memory network, for example, simulating a stack due to the fixed write-one-slot-per-timestep interface.
>
> The purpose of deriving the reductions was to get some insights by comparing neural stack and neural RAM, we didn’t suggest using neural RAM simulating neural stack to solve a problem, and that is the reason why we said “for the tasks where the previous memory needs to be addressed sequentially, the stack neural network is our first choice.”
>
>
> Q4) Nit: I'm not sure the authors should be saying they 'developed' four synthetic tasks, when many of these tasks have previously been proposed and published (counting, copy, reverse copy).
>
> We said ‘developed’ because some of our experiments were not same as before. Our experiments were slightly revised to highlight the advantage and limitation of different memory types. For example, compared to the previous counting task, we added some interferences to the input sequences (see details in our ‘counting with interference’ task). By comparing vRNN’s performance on our ‘counting’ and ‘counting with interference’ task, the limitation of internal memory in vRNN was shown more clearly.  But since our revision is less novel, we have changed ‘develop’ to ‘select’.

---

> > ### Comment · AnonReviewer2 · 2018-11-22
> > **Re. Re.**
> >
> > Thanks for your response and for the updates that you have made to the paper.
> >
> > Q1 response
> > "We agree with your statement and we are sorry for the misleading “single event”. By “single event” we mean that if there is only one useful event, it can be stored as it is, but if there are multiple useful events, they have to be compressed into one compounded event and can only be accessed as a whole. This has been clarified in the text." -
> >
> > Unfortunately I still think this is misleading (for me at least). We have seen over the past two decades that RNNs, and LSTMs in particular, have an amazing ability to store multiple different pieces of information, reason over these disparate chunks and access sub-components. Viewing the single vector state as 'one compound event that can only be accessed as a whole' gives the impression of something more rigid in my opinion.
> >
> > Q2 response
> > Thanks for the response about sentiment analysis! My suspicion is that whenever an LSTM performs better than a MANN this paper can just summarize the reason as an explanation of why the underlying task is a low memory-capacity task. However some tasks like language modelling appear (not sentiment analysis, language modelling) appear as though they would benefit from a longer context and are not implicitly low-memory. Thus I don't feel like this really explains what holds back current MANN architectures (LSTMs and the Transformer always beat them) - I would suspect it is because the optimization for DNCs etc. is more difficult because they have a small number of scalar gates (e.g. write gate) which have a big impact on performance. It would be nice to strive for a taxonomy which can make a prediction about task suitability that is non-obvious (e.g. isn't just "does this task require a lot of memory up-front").
> >
> > Q3)
> > I agree I am not saying that we should use MANNs to simulate neural stacks. I am saying that the reduction is either (a) not possible for some architectures, e.g. a Memory Network, (b) possible but only with significant architectural changes, e.g. a santoro et al. (2016) style MANN, (c) trivial, e.g. a DNC with the temporal linkage matrix, which can implement a stack. Thus I am not sure if it's a valuable reduction.
> >
> > I think the more interesting distinction between memory systems is whether or not the number of trainable parameters is tied to the memory size (as this dictates how much fidelity we have over memory), whether or not the system requires backpropagation-through-time (e.g. MemNet = No, DNC = Yes) for optimization, as this dictates how easy it is to train and whether it will generalize well to longer sequences,  how much information in memory can be modified at any given time-step, ...
> >
> > Unfortunately I still do not feel positive about accepting this paper but I have raised my score.

---

> > > ### Author Response · Authors · 2018-11-27
> > > **Re.Re.Re._part2**
> > >
> > > Q3) I agree I am not saying that we should use MANNs to simulate neural stacks. I am saying that the reduction is either (a) not possible for some architectures, e.g. a Memory Network, (b) possible but only with significant architectural changes, e.g. a santoro et al. (2016) style MANN, (c) trivial, e.g. a DNC with the temporal linkage matrix, which can implement a stack. Thus I am not sure if it's a valuable reduction. I think the more interesting distinction between memory systems is whether or not the number of trainable parameters is tied to the memory size (as this dictates how much fidelity we have over memory), whether or not the system requires backpropagation-through-time (e.g. MemNet = No, DNC = Yes) for optimization, as this dictates how easy it is to train and whether it will generalize well to longer sequences, how much information in memory can be modified at any given time-step, ...
> > >
> > > The reduction from neural RAM to neural stack can not apply to all the architectures (such as memory network). We include it in the paper just for completeness. However, whether this reduction is valid for all types of neural RAM does not affect our basic argument (whether the stored events can be accessed in arbitrary order or not). We will add some comments to make this clear in the final paper.
> > >
> > > The differences between neural RAM and stack can be analyzed in many aspects. Our paper focuses on whether the stored events can be accessed in arbitrary order or not. There are some other inter-class/intra-class differences exists, like what reviewer mentioned, whether the number of parameters equals to the number of memory slots. If a paper’s target is only to compare neural stack and neural RAM then YES, it should be discussed in detail (although we think it’s not hard to infer from our proposed taxonomy). And whether the model is BPTT-free is also an interesting problem, but it is not unique to memory networks. Since the goal of this paper is to propose a unifying framework to analyze the memory structures of these four popular networks, we think these distinctions are not closely related to the goal, although they are very interesting. We may leave them for future work.

---

> > > ### Author Response · Authors · 2018-11-27
> > > **Re.Re.Re_part1**
> > >
> > > Q1) Unfortunately I still think this is misleading (for me at least). We have seen over the past two decades that RNNs, and LSTMs in particular, have an amazing ability to store multiple different pieces of information, reason over these disparate chunks and access sub-components. Viewing the single vector state as 'one compound event that can only be accessed as a whole' gives the impression of something more rigid in my opinion.
> > >
> > > It can be clearly seen from the memory update equations of LSTM,
> > > h_t=o_t*tanh(m_t)  (READ)
> > > m_t=f_t*m_{t-1}+i_t*c_t (WRITE)
> > > When LSTM reads from its external memory m_t, o_t*tanh(m_t) is obtained. Notice that o_t is a scalar. m_t stores ONE compound event (it can also be seen as the state of the network, different from vRNN, this state memory of LSTM has more flexibility due to its gate mechanism), it cannot be read partially. When there is a new event needs to be stored, it has to either be combined with the old one m_{t-1}( 0<f_t<1, 0<i_t<1) or replace the old event ( f_t=0, i_t=1). The problems for which LSTM give good results only use one composite event at a time.
> > > If the reviewer still insists on his/her opinion, please provide some reference paper. We will be so grateful if we know we make a mistake so we can correct it and improve the quality of the paper.
> > >
> > > Q2) Thanks for the response about sentiment analysis! My suspicion is that whenever an LSTM performs better than a MANN this paper can just summarize the reason as an explanation of why the underlying task is a low memory-capacity task. However some tasks like language modelling appear (not sentiment analysis, language modelling) appear as though they would benefit from a longer context and are not implicitly low-memory. Thus I don't feel like this really explains what holds back current MANN architectures (LSTMs and the Transformer always beat them) - I would suspect it is because the optimization for DNCs etc. is more difficult because they have a small number of scalar gates (e.g. write gate) which have a big impact on performance. It would be nice to strive for a taxonomy which can make a prediction about task suitability that is non-obvious (e.g. isn't just "does this task require a lot of memory up-front").
> > >
> > >
> > > First of all, please don’t confuse memory-capacity and how many events can be stored. Memory capacity represents the size of the memory which is related to the number of neurons and parameters. In our paper, we talked about the memory access flexibility, i.e., if the memory can be separated into different sub-blocks to store multiple events. For example, even if one memory network can at most store one event, it’s may still have high memory capacity. (like LSTM, if the size of its external memory slot is large, its memory capacity can be very high).
> > >
> > > For tasks which need one event to be stored, LSTM always performs better than DNC (both of them can store the event and LSTM is easier to train, but the trainability is not the goal of this paper). For high memory-capacity language modeling tasks as the reviewer mentioned, we have to make sure whether it needs multiple events or not. If not, LSTM beats neural RAM makes sense. However, if task needs multiple events, the current neural RAM architectures may still not work well. In this case, the current architectures need to be improved. And with the proposed taxonomy, at least we know we should either revise LSTM to store multiple events (like stacked LSTM) or change the architecture of neural RAM to make it easier to train (other than randomly pick several architectures, test them on the task and compare their relatively error).
> > >
> > > Like what we said in our last reply, “memory is a very abstract concept and the specific memory requirement of a specific task is implicit”. There are some quantifiers of memory such as memory depth, memory resolution, memory capacity, etc., but none of them can really help to differentiate these memory architectures. Hence, we decide to use “how many events can be stored and if the access order is restricted” to quantify the capabilities of different memory networks, which is simple but useful in practice.

---

> ### Comment · Area_Chair1 · 2018-11-30
> **Please consider reviewer1's comments**
>
> Hello R2,
>
> Your review is at odds with R1's (in particular). This is absolutely fine, and I am happy to see you have participated in the discussion/rebuttal process with the authors below. If you could take a moment to read R1's comments and briefly discuss where you disagree and agree, that would be helpful as we need to at least attempt harmonization (but it should not be forced, so it's fine if you agree to disagree).
>
> Best,
> AC

---

### Official Review · AnonReviewer3 · 2018-11-02
**Useful taxomony of memory-based neural network**

**Rating:** 5
**Confidence:** 5

**Review:**

Summary
=========
The paper analyses the taxonomy over memory-based neural networks, in the decreasing order of capacity: Neural RAM to Neural Stack, Neural Stack to LSTM and LSTM to vanilla RNN. The experiments with synthetic and NLP datasets demonstrate the benefits of using models that fit with task types.

Comment
========
Overall, the paper is well written and presents interesting analysis of different memory architectures. However, the contribution is rather limited. The proposed taxonomy is not new. It is a little bit obvious and mentioned before in [1] (Unfortunately, this was not cited in the manuscript). The theorems on inclusion relationship are also obvious and the main contribution of the paper is to formally show that in mathematical forms.  The experiments on synthetic tasks give some insights into the models’ operations, yet similar analyses can be found in [2, 3]. To verify the models really learn the task, the authors should include tests on unseen sequence lengths.  There remains questions unexplained in NLP tasks such as why multi-slot memory did not show more advantages in Movie Review and why Neural Stack performed worse than LSTM in bAbI data.

Minor potential errors:

In Eq. (6), r_{t-1} should be r_t

The LSTM presented in Section 3.2 is not the common one. Normally, there should be x_t term in Eq. (3) and h_t=g_{o,t}*\tanh(r_t) in Eq. (6). The author should follow the common LSTM formulas (which may lead to different proofs) or include reference to their LSTM version.

[1] Yogatama et al. Memory Architectures in Recurrent Neural Network Language Models. ICLR’18

[2] Joulin et al. Inferring algorithmic patterns with stack-augmented recurrent nets. NIPS’15

[3] Graves et al. Neural Turing Machines. arXiv preprint arXiv:1410.5401 (2014).

---

> ### Author Response · Authors · 2018-11-22
> **Reply to AnonReviewer3_part1**
>
> Q1) The proposed taxonomy is not new. It is a little bit obvious and mentioned before in [1] (Unfortunately, this was not cited in the manuscript). The theorems on inclusion relationship are also obvious and the main contribution of the paper is to formally show that in mathematical forms
>
> Thank you for mentioning [1] (we have cited it in the revised version for completeness) but we disagree that “the proposed taxonomy is not new”. The authors in [1] simply divided these models into sequential, random access and stack memory architectures, which bears some similarity with the taxonomy proposed in our paper, but it is more superficial and does not go to the mechanisms behind the memory types. Indeed, classifying models according to the type of memory seems obvious, but finding the essential relationship between classes and linking the descriptive power of learning machines to the properties of task data is not a trivial work. (For example, what’s the difference between internal and external memory and what kind of tasks can they address? Our taxonomy showed clearly that the gate mechanism in LSTM and the push/pop/no-op operators in the stack augmented memory had the same function in nature, which had never been mentioned before. Many discussions like these first appeared in our paper.) Many papers proposed fancy models to improve the existing work, however there is still no paper providing a good approach to analyze what the memory architectures can learn and how to select the most parsimonious memory model for a specific task. As far as we know, our paper addresses for the first time how to codify and exploit the knowledge gained from the different characteristics of the memory within the taxonomy to help users select the type of memory network for an application.  Moreover, the effectiveness of this analysis framework was also verified in some simple experiments. However, we agree that the analysis is not complete because on the one hand people has to analyze memory requirements of a task by themselves which is not trivial and on the other hand, the accuracy is also affected by the size and specific network structure even within a class. But we firmly believe that classifying architectures into these four classes and linking the architecture of the learning machine to its descriptive power, as we did in this paper, is a fundamental first step, and we think this paper is important to the machine learning community. We admit that the proofs of the theorems are not very hard and we included them to make our argument rigorous.
>
> Q2) The experiments on synthetic tasks give some insights into the models’ operations, yet similar analyses can be found in [2, 3].
>
> Although [2][3] and our paper used similar synthetic tasks in experiments, our goal was again very different. In [2][3], their goal was to show the effectiveness of their proposed architecture, so each of them only focused on analyzing the operation of one specific model. However, since our goal is to verify the proposed taxonomy, our experiment focused on showing the connections between different memory types and the growing capability of these four classes of models. For example, although [2] showed the details of the operation of neural stack and the neural stack performing better than LSTM on some tasks, we believe readers still won’t understand what was the connection between LSTM and neural stack and why LSTM could be seen as a special case of neural stack. However, in our “counting with interference” task, the results showed that (Appendix D.2) content of the top element of the stack (M0 in Fig.12) had the same changing trend as the external memory of LSTM (M0 in Fig.11) and other content in the stack below the top one was redundant. Hence it helped verify our argument “LSTM can be seen as neural stack with only the top element”. Because of page limitation, we didn’t show how the three gates in LSTM related to the push/pop/no-op operators in neural stack, but our argument would be more convincing if these operator comparison results were added.
> We struggled to demonstrate the capabilities of each memory architecture, and our decision was to construct four representative tasks that would fit optimally the characteristics of each memory organization. Therefore, these four representative tasks were carefully selected to allow practitioners to compare their own problem with these four tasks and give them some hints to select the right model. This has been better explained in the revised paper, but the point is that we are not just simply repeating some existing experiments.

---

> > ### Comment · AnonReviewer3 · 2018-11-26
> > **Reply to response**
> >
> > Thanks for detailed response. Having read the response and other reviews, I have raised my rating to 5. The work is definitely helpful to practitioners, and should be pursued further to the point of suggesting future venues of research.

---

> > > ### Comment · Area_Chair1 · 2018-11-30
> > > **Assessment**
> > >
> > > Thank you for participating in the discussion with the authors, Reviewer 3. Could you please clarify, in your revised assessment, given the discussion you have had with the authors, where the paper still falls short so as to merit a score of 5?
> > >
> > > Additionally, there is a wide spread of scores for this paper. Could you please consider the reviews provided by R1 and R2 and see if there is anything you agree or disagree with in their assessments (and if so, please comment or discuss). There is no requirement that scores be harmonized, but you must ideally at least show consideration for the opinions of the other reviewers.
> > >
> > > Best,
> > > AC

---

> ### Author Response · Authors · 2018-11-22
> **Reply to AnonReviewer3_part2**
>
> Q3) To verify the models really learn the task, the authors should include tests on unseen sequence lengths.
>
> We have these results and will include a table showing the accuracy for each model on longer sequence lengths in the revised version.
>
> Q4) There remains questions unexplained in NLP tasks such as why multi-slot memory did not show more advantages in Movie Review and why Neural Stack performed worse than LSTM in bAbI data.
>
> From our observation of the results, all the models solved the sentiment analysis problem mainly based on some discriminating words. Specifically, when feeding the input sequence to the model, the output value would be increased when meeting positive words such as “good, love” and decreased when meeting negative words such as “dislike, boring”. If there were many positive words appearing in text, the sentiment would be judged as positive. Hence, we only need an external memory whose value can be affected by the discriminating words. Although multi-slot memory can store more than one event, as long as it cannot understand the logic of the text, its performance cannot be improved compared to the model with single slot memory(LSTM). Hence the multiple slots memory didn’t not show more advantage compared to the single slot memory.
>
> For the bAbI data, in order to solve the problem, learning machines need to store all the potential useful facts and read any of them when needed, so a multi-slot external memory whose contents can be randomly accessed is necessary. Hence, both LSTM and neural stack are not suitable for this task. If we apply LSTM or neural stack to the problem, they will try their best to find their approximate solutions. If we apply LSTM to this task, LSTM would compress all the useful information in its external memory, although the right answer is mixed with other information, the output can at least get some information from it. However, if we apply neural stack to this task, the push signal sometimes is very large (d_push~=1, d_pop~=0, d_no_op~=0), which means that the right answer will then be pushed down in the stack which cannot be accessed when needed. Hence, we think although the neural stack tries its best to find the right answer, its more complicated operation may make it is more likely to be stuck in local points. This has been mentioned in the revised manuscript.
>
> Q5) Minor potential errors: In Eq. (6), r_{t-1} should be r_t.
>
> Actually, it’s not an error. Since in Eq.(3), c_t is a function of h_t, it should be r_{t-1} in Eq(6). But we will change r_{t-1} to r_t in Eq.(6) and h_t to h_{t-1} in Eq.(3) if this form is more formal.
>
> Q6) The LSTM presented in Section 3.2 is not the common one. Normally, there should be x_t term in Eq. (3) and h_t=g_{o,t}*\tanh(r_t) in Eq. (6). The author should follow the common LSTM formulas (which may lead to different proofs) or include reference to their LSTM version.
>
> When the models discussed in our paper were first proposed, their dynamical equations look very different (e.g., in LSTM, c_t is a function of h_{t-1} and x_t; in neural stack, c_t is a function of h_{t-1}). Since the goal of our paper is to analyze the connection and difference between different models, it’s better to use uniform dynamical equations to describe the memory system. In this way, it’s much easier to see their essential differences. Hence, we used the revised version of LSTM in our paper since it doesn’t affect the basic working mechanism of the architecture.

---

### Official Review · AnonReviewer1 · 2018-11-03
**Very interesting consolidation paper on the analysis of dynamic neural networks**

**Rating:** 7
**Confidence:** 3

**Review:**

I really liked this paper and believe it could be useful to many practitioners of NLP, conversational ML and sequential learning who may find themselves somewhat lost in the ever-expanding field of dynamic neural networks.

Although the format of the paper is seemingly unusual (it may feel like reading a survey at first), the authors propose a concise and pedagogical presentation of Jordan Networks, LSTM, Neural Stacks and Neural RAMs while drawing connections between these different model families.

The cornerstone of the analysis of the paper resides in the taxonomy presented in Figure 5 which, I believe, should be presented on the front page of the paper. The taxonomy is justified by a thorough theoretical analysis which may be found in appendix.

The authors put the taxonomy to use on synthetic and real data sets. Although the data set taxonomy is less novel it is indeed insightful to go back to a classification of grammatical complexity and structure so as to enable a clearer thinking about sequential learning tasks.

An analysis of sentiment analysis and question answering task is conducted which relates the properties of sequences in those datasets to the neural network taxonomy the authors devised. In each experiment, the choice of NN recommended by the taxonomy gives the best performance among the other elements presented in the taxonomy.

Strength:
o) The paper is thorough and the appendix presents all experiments in detail.
o) The taxonomy is clearly a novel valuable contribution.
o) The survey aspect of the paper is also a strength as it consolidates the reader's understanding of the families of dynamic NNs under consideration.

Weaknesses:
o) The taxonomy presented in the paper relies on an analysis of what the architectures can do, not what they can learn. I believe the authors should acknowledge that the presence of Long Range Dependence in sequences is still hard to capture by dynamic neural networks (in particular RNNs) and that alternate analysis have been proposed to understand the impact of the presence of such Long Range Dependence in the data on sequential learning. I believe that mentioning this issue along with older (http://ai.dinfo.unifi.it/paolo/ps/tnn-94-gradient.pdf) and more recent (e.g. http://proceedings.mlr.press/v84/belletti18a/belletti18a.pdf and https://arxiv.org/pdf/1803.00144.pdf) papers on the topic is necessary for the paper to present a holistic view of the matter at hand.
o) The arguments given in 5.2 are not most convincing and could benefit from a more thorough exposition, in particular for the sentiment analysis task. It is not clear enough in my view that it is true that "since the goal is to classify the emotional tone as either 1 or 0, the specific contents of the text are not very important here". One could argue that a single word in a sentence can change its meaning and sentiment.
o) The written could be more polished.

As a practitioner using RNNs daily I find this paper exciting as an attempt to conceptualize both data set properties and dynamic neural network families. I believe that the authors should address the shortcomings I think hinder the paper's arguments and exposition of pre-existing work on the analysis of dynamic neural networks.

---

> ### Author Response · Authors · 2018-11-22
> **Reply to AnonReviewer1**
>
> Q1) The taxonomy presented in the paper relies on an analysis of what the architectures can do, not what they can learn. I believe the authors should acknowledge that the presence of Long Range Dependence in sequences is still hard to capture by dynamic neural networks (in particular RNNs) and that alternate analysis have been proposed to understand the impact of the presence of such Long Range Dependence in the data on sequential learning. I believe that mentioning this issue along with older (http://ai.dinfo.unifi.it/paolo/ps/tnn-94-gradient.pdf) and more recent (e.g. http://proceedings.mlr.press/v84/belletti18a/belletti18a.pdf and https://arxiv.org/pdf/1803.00144.pdf) papers on the topic is necessary for the paper to present a holistic view of the matter at hand.
>
> We agree that the LRD problem is hard for RNNs, and this is the major reason external memories are needed, In the revised version, we have motivated the need for the taxonomy by the LRD problem and included these three papers for completeness. However, we would like to say that the main goal of our paper is indeed to explain what each architecture can learn from data, against your first observation. Based on this we further analyzed what they can do. From your comment we may have emphasized too much the aspect of how they can be used to help the practitioner. We have characterized what information each architecture can extract from the data stream in the revised manuscript.
>
> Q2) The arguments given in 5.2 are not most convincing and could benefit from a more thorough exposition, in particular for the sentiment analysis task. It is not clear enough in my view that it is true that "since the goal is to classify the emotional tone as either 1 or 0, the specific contents of the text are not very important here". One could argue that a single word in a sentence can change its meaning and sentiment.
>
> From our analysis of the results, all the models solve the sentiment analysis problem mainly based on the occurrence and reoccurrence of some discriminating words. Specifically, when feeding the input sequence to the model, the output value would be increased when meeting positive words such as “good, love” and decreased when meeting negative words such as “dislike, boring”. If there were many positive words appearing in text, the sentiment would be judged as positive.  Since the model only cares about whether the word is positive or negative and its number of occurrences in the text (kind of a “density”), we deduct that a specific word is not crucial (for example, as long as it is a positive word, whether it is “love”,’like’ or ‘happy’ is not that important), and translated this as “the specific contents of the text are not very important”. We have elaborated more on this point in the final version because as the reviewer pointed out the explanation is too brief and not specific.
>
> But we have to be aware that this “discriminating words based” method does not really solve the problem as human. As the reviewer mentioned “a single word in a sentence can change its meaning and sentiment”, therefore, in order to really solve this problem, the machine should learn how to decode the text meaning. But none of the current models can really achieve this. Although these models can capture some temporal dependencies (for example, if there is a “don’t” before “like”, the sentence is more likely to be negative), the final decision still mostly depends on how many “discriminating words” appear in the text. That’s also the reason why they cannot get 100% accuracy.
>
> Q3) The written could be more polished.
> We have further polished the language.

---

> ### Comment · Area_Chair1 · 2018-11-30
> **Discuss with other reviewers**
>
> Dear Reviewer 1,
>
> Thank you for your detailed review. Your score is at odds with the other reviewers, which is absolutely fine! I would appreciate if you could take a minute to consider author comments and the other reviews. If you are willing to champion the paper, which you seem to be given your score, please attempt to convince the other reviewers of your viewpoint. Alternatively, if you see merit in their concerns, please adjust your assessment accordingly.
>
> It is okay to agree to disagree at the end of this process, but there must be some attempt at harmonization.
>
> Thanks!
>
> AC

---

> > ### Comment · AnonReviewer1 · 2018-11-30
> > **Revised assessment: lowered score and confidence based on discusssions**
> >
> > The discussions with the two other reviewers have shown me that my enthusiasm for the matter at hand may have clouded my judgement.
> >
> > Although I maintain that the paper may be helpful for practitioners, in particular because of its identifying and comparing different kinds of architectures, the other reviewers do see the lack of clear novelty, the writing and some over-simplifications as important issues. Therefore, although I still believe the paper to be a good paper, I was probably wrong in my first assessment and changed it based on the elements presented in the discussions.

---

> > > ### Comment · Area_Chair1 · 2018-11-30
> > > **Revised assessment**
> > >
> > > You are welcome to adjust your score if you think it reflects your understanding of the paper's strength, but please be reminded there is absolutely no need to agree with other reviewers, or reconcile scores. If you think it's worth a high score, you are more than welcome to keep it like that: you just ideally need to provide further justification for your position to enable the AC and PC to make an informed decision.

---

### Meta-Review · Area_Chair1 · 2018-12-13
**Borderline**

**Confidence:** 4
**Recommendation:** Reject

**Metareview:**

This paper presents a taxonomic study of neural network architectures, focussing on those which seek to map onto different part of the hierarchy of models of computation (DFAs, PDAs, etc). The paper splits between defining the taxonomy and comparing its elements on synthetic and "NLP" tasks (in fact, babi, which is also synthetic). I'm a fairly biased assessor of this sort of paper, as I generally like this topical area and think there is a need for more work of this nature in our field. I welcome, and believe the CFP calls for, papers like this ("learning representations of outputs or [structured] states", "theoretical issues in deep learning")). However, despite my personal enthusiasm, the reviews tell a different story.

The scores for this paper are all over the place, and that's after some attempt at harmonisation! I am satisfied that the authors have had a fair shot at defending their paper and that the reviewers have engaged with the discussion process. I'm afraid the emerging consensus still seems to be in favour of rejection. Despite my own views, I'm not comfortable bumping it up into acceptance territory on the basis of this assessment. Reviewer 1 is the only enthusiastic proponent of the paper, but their statement of support for the paper has done little to sway the others. The arguments by reviewer 3 specifically are quite salient: it is important to seek informative and useful taxonomies of the sort presented in this work, but they must have practical utility. From reading the paper, I share some of this reviewer's concerns: while it is clear to me what use there is the production of studies of the sort presented in this paper, it is not immediately clear what the utility of *this* study is. Would I, practically speaking, be able to make an informed choice as to what model class to attempt for a problem that wouldn't be indistinguishable from common approaches (e.g. "start simple, add complexity"). I am afraid I agree with this reviewer that I would not.

My conclusion is that there is not a strong consensus for accepting the paper. While I wouldn't mind seeing this work presented at the conference, but due to the competitive nature of the paper selection process, I'm afraid the line must be drawn somewhere. I do look forward to re-reading this paper after the authors have had a chance to improve and expand upon it.